# Evaluating the Bovine Tuberculosis Eradication Mechanism and Its Risk Factors in England’s Cattle Farms

**DOI:** 10.3390/ijerph18073451

**Published:** 2021-03-26

**Authors:** Tabassom Sedighi, Liz Varga

**Affiliations:** 1Centre for Environmental and Agricultural Informatics, School of Water, Energy and Environment (SWEE), Cranfield University, Cranfield MK43 0AL, UK; 2Department of Civil, Environmental and Geomatic Engineering, Faculty of Engineering, UCL, London WC1E 6BT, UK; l.varga@ucl.ac.uk

**Keywords:** evaluation, infectious disease modelling, dynamic bayesian network, bovine tuberculosis, sensitivity analysis, planning

## Abstract

Controlling bovine tuberculosis (bTB) disease in cattle farms in England is seen as a challenge for farmers, animal health, environment and policy-makers. The difficulty in diagnosis and controlling bTB comes from a variety of factors: the lack of an accurate diagnostic test which is higher in specificity than the currently available skin test; isolation periods for purchased cattle; and the density of active badgers, especially in high-risk areas. In this paper, to enable the complex evaluation of bTB disease, a dynamic Bayesian network (DBN) is designed with the help of domain experts and available historical data. A significant advantage of this approach is that it represents bTB as a dynamic process that evolves periodically, capturing the actual experience of testing and infection over time. Moreover, the model demonstrates the influence of particular risk factors upon the risk of bTB breakdown in cattle farms.

## 1. Introduction

The bTB is an incessant bacterial disease of cattle caused by M. bovis (Mycobacterium bovis) infection. The disease is a persistent economic and veterinary problem in cattle herds in the UK despite the policy of cattle herd testing and slaughter [1]. bTB affects animal health and welfare, causes financial strain through involuntary culling and animal movement restrictions and incurs costs through control and eradication programs [2,3].

In England, the majority of bTB cases are reported in southwestern England and Wales which are considered to be high-risk areas. To control and annihilate bTB in these areas, a control program has been in place since 1947 consisting primarily of routine and targeted surveillance of cattle herds, culling of positively tested for bTB animals and movement restrictions on infected herds [4].

Surveillance is based on the administration of a single intradermal comparative cervical tuberculin test (skin test) involving two separate injections of sterile purified mixtures of Mycobacterium avium and M. bovis antigens (tuberculins) in the deep layer of the skin of the neck, followed by examination of the skin for localised allergic reactions after 72 h [5,6].

For negative or non-reactor skin test results, the reaction to the M. bovis tuberculin injection is reported less than or equal to that to the M. avium tuberculin injection, while a positive skin test result (known as a reactor) is declared when the reaction to M. bovis tuberculin exceeds that of M. avium tuberculin by more than 4 mm, according to the standard international interpretation [5]. In all other cases, the test is considered inconclusive and is redone again in two months time. A new herd incident (NHI) or breakdown is declared when one or more animals in a herd have positive skin test results. The breakdown in a herd causes immediate animal movement limits, tentative termination of the official bTB free (OTF) status of the herd and a retest of all of the remaining animals in the herd at two-month intervals [7].

Animals with a positive or two inconclusive skin test results in sequence are immediately slaughtered and tested at the slaughterhouse for noticeable lesions of bTB in their organs. When the result of the postmortem examination is positive because of the presence of lesions, positive M. bovis culture, or both, then the herd’s status from suspended downgrades to withdrawn [7]. Infection of cattle with M. bovis is usually chronic and can remain subclinical for a long period. Importantly, infected cattle can become infectious long before they exhibit any obvious clinical signs or lesions typical of bTB detectable even with the most careful veterinary examination [5]. Thus, if there is a breakdown and the status is withdrawn, then the herd examination continues on all remaining cattle until the results of the skin tests on the herd after one or two (based on the risk areas which the herd is in and the postmortem results) consecutive tests become negative at minimum intervals of two months [7].

Although lots of efforts and investments have been made to prevent NHI and herd breakdowns in England, they constantly increased between the mid-1980s and 2012 [7]. However, the test and slaughter programme remains pivotal despite the increase in the bTB cases [4], and the number of cattle herds placed under movement restrictions in England due to the suspected presence of bTB has progressively increased over the past decades [8].

The bTB epidemiology, in England, is complicated, and it is difficult to define the relationship between evidence, uncertainty and risks [9,10]. Moreover, the control of bTB in England is complicated by the involvement of wildlife, particularly badgers, which appear to sustain endemic infection and can transmit bTB to cattle. Badgers are implicated as an important wildlife reservoir and an obstacle to reducing bTB incidence [11,12,13].

Despite the non-husbandry risk factors which are related to the environmental factors such as badger sett density and habitat composition, the husbandry and farm management-related risk factors also contribute to the heightened risk of herd breakdown [14,15]. The perceived risks relate to: (i) environmental biosecurity including farm waste management and foodstuff storage [16]; (ii) contact between cattle and badgers or their excreta at pasture [17]; (iii) herd characteristics such as herd size, farm enterprise and animal movements [18,19]; and (iv) animal management including stocking densities, feeding and grazing regimes [20,21].

Thus, the identification of variables which are statistically significantly affecting bTB status in England is essential and important to control bTB [22]. For instance, in [23], a pipeline for the prediction of bTB status in dairy cows by applying state-of-the-art deep-learning techniques to their milk MIR spectral profiles is presented. Moreover, in [24], the related steps of the machine learning techniques for the predictive analysis of bovine tuberculosis in cattle from the state of Sao Paulo, Brazil is given. Therefore, it is important to describe a model for quantifying the effect of the considered risk factors of England’s cattle farms in different risk areas.

In the last few years, Bayesian Network (BN) has rapidly been adopted across different areas of science [25,26,27,28]. A BN as a probability-based knowledge representation method is appropriate for the modelling of causal processes with uncertainty [29,30]. The BN is a graphical decision tool with nodes representing the states of the system and links encapsulating the evidence for dependence between parent and child nodes. In our presented BN model, all risk factors related to bTB are considered as parent nodes and all evidence related to the breakdown statuses as the child nodes. Any other node in our model such as the skin test results and the movement restriction (hidden nodes) can be both child and parent nodes. A BN uses bidirectional probabilistic reasoning and is able to handle the uncertainty throughout the model [31], which provides insight when we know something about the relationships between variables. In our model, uncertainty arises from the inability to evaluate the degree of truth of a hypothesis due to unreliable and incomplete information or inconsistent knowledge [32]. Moreover, when you want to model a domain in a way that is visually transparent, as well as aim to capture cause–effect relationships, BNs can be very powerful.

For example, Smith et al. [31] explored the use of a BN to model ecosystem services and Carriger et al. [33] highlighted opportunities for using Bayesian tools to combine evidence during ecological risk assessment. They demonstrated how probabilistic and causal reasoning about complicated problems are accommodated in BNs and discussed how a causal BN approach would benefit future evidence synthesis in environmental management and risk assessment by improving inferences.

Moreover, most of the methods in policy and evaluation studies are usually based on statistical tests which are performed on functional or structural analysis across or within subjects. Such methods are definite in nature and they have challenges when there is a lack of good-quality data. In this paper, we propose a DBN approach combining the data and expert knowledge and other reliable information to evaluate the consequences of the existing bTB prevention mechanisms in the England’s cattle farms. Our application of such procedure describes nonlinear multivariate relationships between the England’s cattle farms husbandry structure, Defra’s test and slaughtering policy and other measurements of interest.

This manuscript is organised as follows. Section 2 and Section 3 provide a discussion on BNs, our case study, bTB and its DBN model, respectively. Section 4 studies the data and statistical procedures. The simulation results and the discussions are presented in Section 5, while the conclusions are provided in Section 6.

## 2. Bayesian Network

In general, probabilistic models are more complicated than deterministic ones. It means that they need more knowledge and attention to the structure and sensible choices of data or input distributions are very important. When the available data are not appropriate, more challenges will arise in terms of uncertainty [34]. A BN is a probabilistic graphical model which is used to represent knowledge about an uncertain domain [35]. Each random variable (a variable whose possible values are outcomes of a random phenomenon) is represented by a node in the BN. A conditional probability table is attached to each node. A link (or ‘edge’) between two nodes represents a probabilistic dependency between the linked nodes. The links are shown with an arrow pointing from the causal node to the effect node (the links are ‘directed’). There must not be any directed cycles: one cannot return to a node simply by following a series of directed links. This means that BNs are Directed Acyclic Graphs (DAGs). Nodes without a child node are called leaf nodes, nodes without a parent node are called root nodes and nodes with parent and child nodes are called intermediate nodes. A BN represents dependence and conditional independence relationships among the nodes using joint probability distributions, with the ability to incorporate human oriented qualitative inputs [36]. The method is well established for representing cause–effect relationships (Figure 1).

Deeper understanding of dependencies and relationships between variables can be achieved through more investigation in Bayesian statistics [37].

To construct a BN, three steps are required. First, determine the Directed Acyclic Graph [38]. Next, assign the prior probabilities of each variable based on measurements, human experts and any other reliable sources. Last, find the posterior probabilities of each variable acknowledging the nodes’ connections. The posterior probabilities are the reason for using BNs. These provide the likelihood of cause–effect relationships on nodes in the system.

There are different types of BNs:Static BN (SBN): All variables are discrete.Dynamic BN (DBN): It consists of a limited number of BNs, each of which corresponds to a particular time interval. The connections between adjacent BNs represent how the states of the system evolve over time. Figure 2 shows a simple two-node DBN with feedback over four time steps.In Figure 2, node A at t1 affects node B at current time slice t1, but it is in turn affected by node B in the next time slice, t2. This amounts to a feedback loop, since A affects B and B affects A (at time t2).DBN is a powerful algorithm that can be applied to various applications with the temporal dependencies in sequential data. Moreover, inference and learning with DBNs including temporal structural learning and prediction have been used routinely in several fields where the filtering and smoothing are performed in Kalman filter models [39].Hybrid dynamic BN (HDBN): Random variables can be discrete such as someone’s gender or continuous such as someone’s age. A BN which contains both discrete and continuous variables is called an HDBN.

Unlike Naïve BN, which only considers conditionally dependent relationships among the random variables by a direct link between those variables, BNs can capture both the conditionally dependent relationships (direct cause and effect relationships) and conditionally independent relationships (the variables which are not connected by a direct link). The probability of variables which are conditionally dependent are obtained where the probability of their parents are known and the probability of the conditionally independent variables are independent of each other when another “separating” variable is known (e.g., a joint parent).

## 3. Case Study: Dynamic Bayesian Network for bTB in the England’s Cattle Farms

We use a DBN to identify risk factors in England’s cattle farms in spreading bTB [40]. We explore the relationships between different risk factors of England’s cattle farms and their roles individually in spreading bTB, particularly to identify those which have the highest impact. We use the network, which is a computational model, to examine the effect of current management practices in cattle farms upon NHI.

### 3.1. Bovine Tuberculosis

bTB is considered an ongoing problem in cattle farms in England despite current policy including cattle herd testing and slaughter. Besides badgers acting as an important wildlife reservoir, cattle husbandry and farm management practices play a role in heightening the risk of spreading the disease. The following definitions are essential for this report:**Failed test**: At least one animal tests positive during any herd-level test (routine, whole-herd or follow up).**Breakdown**: A failed test occurs on an officially bTB free herd, not currently subject to movement restriction.

### 3.2. Dynamic Bayesian Network

DBNs are used generally to explore and display causal relationships, between key factors and outcomes in a complex system, in a straightforward and understandable manner. Moreover, DBNs can also be used to calculate the effectiveness of interventions, such as alternative management decisions or policies where the associated uncertainties with these causal relationships can also be explored at the same time.

### 3.3. Method: Identification of Variables and Their States

#### Farm’s Risk Factors

There are several risk factors related to the bTB disease: (i) environmental biosecurity such as farm waste and management and foodstuff storage; (ii) herd characteristics such as herd size, farm enterprise and animal movements; (iii) contact between cattle and badgers; and (iv) animal management including stocking densities, isolation, feeding and grazing regimes. Among all of these key factors, the following are considered in our model:**Farm food**: There are different types of food and storage methods and storage periods. However, routes of transmission from badgers that are attracted to the cattle food are considered important. Studies show that farms that feed maize silage, grass silage or molasses to cattle are more attractive to badgers. In addition, studies show that 100% of farms with persistent breakdown fed grass silage to cattle, whereas feeding hay was effective against persistent breakdown.**Badgers**: Active badger sett density showed significant associations with herd incidents. In our model, active sett density was entered as a categorical variable as zero, medium (≤0–3 setts/km) or high (>3 setts/km). Based on available studies, a relatively high active sett density is associated with persistent breakdown, which shows that badgers play a part in spreading the disease, especially in hotspot areas.**Age of the cattle**: Based on available information, age of purchased animals was categorised as:–Calves: Stores or replacement breeding animal less than 12 months–Yearlings: Stores or replacement breeding animal between 12 and 24 months–Cows: Female breeding animals over 24 monthsBased on available information, farms with more cattle older than 24 months are more likely to find bTB becomes persistent compared with farms with younger cattle. In other words, bTB prevalence tends to increase with age.**Stock density**: Based on available data, farms with relatively low stocking densities (≤3 head of cattle/ha) were more protected from breakdown versus the farms with high densities (>3 head of cattle/ha).**Purchased cattle**: In control farms, the main reason of increasing the risk of infection is cattle movement onto farms from market or other farms. Since purchased cattle is responsible for the majority of newly infected farms, buying animals accelerates bTB incidents. Studies demonstrate that farms with the purchase of larger number of cattle (>50 heads compared to ≤50 heads) are at more risk of breakdown (short period is more likely than persistent).**Manure storage**: Based on available literature, manure storage for over six months is one of the significant factors for breakdown in the farms.

### 3.4. Data and Statistical Procedures

Data are available publicly as secondary data in Quarterly TB in cattle in Great Britain statistical notice (data to March 2018). Data are available for Great Britain, England, high-risk, edge-risk and low-risk areas of England, Wales and Scotland. Data are mostly presented as monthly data or yearly data from 1996 to 2016. Some of the data are available at herd level such as total test on herds, NHI, not officially TB-free herds and herds under movement restriction. Some data are presented at a number of animal levels such as total number of slaughtered animals, total number of animals reported to the slaughter house and the number of confirmed slaughtered. Our model is a herd-level model that does not distinguish between individual cattle on a farm, thus we include herd-level data but not individual cattle data.

### 3.5. Dynamic Bayesian Network Model

Our DBN model allows analysts to make probability judgments consistent with direction of causality [40]. Since DBNs are able to work with data of different types and sources, they handle a mix of subjective and objective data and, hence, supplement traditional experimental and statistical methods. Moreover, DBN models can update hypothesis probabilities with new information, such that the effects of one month or time period are carried forward to have effects upon events in future time periods. Thus, DBNs can evaluate the reliability of results when new evidence is added during the analysis period.

#### 3.5.1. Nodes and Edges

In our model, nodes are elicited from the literature and available documents regarding the bTB policy published by the government. The list of all nodes and their states are presented in Table 1 and Table 2. Edges, which show the level of effectiveness between the different nodes as well as the causal relationships between them, are elicited from the domain experts and the available literature (see Figure 3).

#### 3.5.2. Prior Probabilities

The prior probability of an uncertain quantity is the probability distribution that would express one’s belief about this quantity before some evidence is taken into account [41]. In our model, prior probabilities assigned to root nodes and conditional probabilities (probability of each value of a node given the value of its parent) for leaf nodes are elicited from the literature, expert domain and available data [42].

This project was supported by the Centre for the Evaluation of Complexity Across the Nexus (CECAN), which comprises social scientists, policy makers, policy analysts and other academic researchers who have collaborated in expert elicitation. In this research, the main source of information was retrieved from a comprehensive literature review, which was used to train the BN structure and estimate the conditional probabilities. The learned BN was then revised by the domain experts, including experts in the fields of sociology and complex systems to avoid the cognitive factors and biases that could affect the aggregated elicited judgments [43].

#### 3.5.3. Interslice Probabilities

Dynamic networks are merely static networks straddling time periods or time slices. The future state of some nodes can be influenced by their prior state (feedback nodes). Some nodes can also effect other nodes in other time slices. The prior probabilities of these nodes are also assigned before evidence is added to the model. In Figure 4, Node 7 is feeding another node in a different time slice.

#### 3.5.4. Posterior Probabilities

The final step in the model is to run the DBN at monthly intervals. For each time slice, any new information is added to the model to update current priors. All updated probabilities from the first to the last time slice are presented as time dependent figures.

## 4. Simulation Results

A combination of the data-driven and knowledge-based approaches was utilised in our study for the construction and evaluation of the proposed DBN model (the DBN structure and its prior probabilities) to identify the main risk factors and the severity rate of NHI of cattle farms in high-risk areas of England. Then, the prior probability distribution of the variables is updated when a new observation of the interested variables is made to show different scenarios. The updated or posterior distributions of all the other variables are then computed by using the inference algorithms. The emerging risk factors will be detected using the bottom-up inference in the proposed DBN [40].

These probabilities explore the relationships between different risk factors and their roles individually in spreading bTB, particularly to identify those which have the highest impact. We then use the top-to-bottom inference to examine the effect of current management practices in cattle farms upon new herd bTB incidents and provide a measure of the severity of the new herd bTB incident or the breakdowns as a set of probabilities.

Outcomes are then given to show the applicability of this method to diagnose the high-impact risk-causes in the England’s cattle farms with the current Defra policy on bTB.

The following steps were performed in order to achieve the presented results:After the DBN graph was constructed and the prior probabilities were elicited from the available literature and data, the DBN updated the prior probabilities every time new information (evidence) was added to the network.Each node has upper, estimated and lower bounds. Thus, the proposed DBN was run several times to compute the posterior probabilities of the interested variables for all three bounds of each node and different node combinations (different scenarios). This computation allows us to handle the uncertainty due to the lack of information in selecting our priors (see Table 3, Table 4, Table 5, Table 6, Table 7, Table 8, Table 9, Table 10 and Table 11).The DBN ran 66 times for each risk area (198 times in total) to define the sensitivity sets for each area. The sensitivity sets are the risk factors which have the greatest influence on NHI.Finally, the DBN ran for different combinations of sensitivity sets to find out which risk factors may have grater effect if they are combined together.

The following figures demonstrate the effectiveness of our DBN to identify the effect of each risk factor and their combinations on NHI.

Figure 5 shows the probabilities of each root node when new data for officially TB free status withdrawn (OTW), as evidence, are added to the DBN. This figure demonstrates that the probability of the silage fed cattle being the cause of OTW was constant for the whole period of 2008–2015. However, the probability that manure storage over six months is the cause of OTW was increased and has a greater effect (see Figure 6).

Moreover, Figure 7 shows the probabilities of each root node when new data for officially TB free status unclassified (OTUC), as evidence, are added to the DBN. This figure demonstrates that:The probability of silage being the cause of OTUC decreased.The probability of the purchased cattle over 50 heads being the cause of OTUC decreased.Stock density and manure storage over six months had a greater chance to be the causes of OUTC.

Figure 8 also shows the probabilities of each root node when new data for officially TB free status suspended (OTS), as evidence, are added to the DBN. This figure demonstrates that:Silage had a greater chance to be the cause of OTS (in comparison with OTW and OTUC).The effect of purchased cattle over 50 heads was decreasing, although its effect was greater than the age of cattle.Stock density and manure storage had the greater chance to be the causes of OTS with some level of uncertainty by the end of the simulation.

At this point, we used the bottom-up inference outcomes to identify our sensitivity set. Figure 9 shows the effect of each node on the OTW, OTUC and OTS probability from 2008 to 2015 respectively. This figure summarises the findings in Table 3, Table 4, Table 5, Table 6, Table 7, Table 8, Table 9, Table 10 and Table 11 to identify the sensitivity sets for the high-risk areas.

Figure 9 shows that the sensitivity set after 66 times of repeated simulation for different probability combinations was identified as SS = [Age, Badgers, Silage].

As shown in Figure 10, Figure 11, Figure 12, Figure 13, Figure 14, Figure 15, Figure 16, Figure 17 and Figure 18, we used the up-bottom inference of our DBN model to measure the effect of the defined sensitivity set in different combinations on the NHI. Figure 10 shows the effect of young cattle (under 24 months) and silage fed on the NHI. This figure shows that young silage fed cattle can decrease the OTW and OTUC.

Figure 11 shows that the probability of NHI was increasing significantly in the presence of badgers and silage fed cattle at the same time.

Moreover, Figure 12 and Figure 13 show that in the presence of badgers and silage fed, the age of cattle is not important and the effect of age on NHI is not impressive, even though age was one of the sensitivity set factors.

Figure 14 and Figure 15 also demonstrate that the effect of manure storage on NHI in the presence of badgers and silage fed was not effective.

However, Figure 16 shows that, in the absence of badgers and silage fed and presence of all other factors, the probability of NHI is decreasing significantly. In other words, if the badger density were 0–3 setts/ha and the cattle were not silage fed, but instead the age of cattle were over 24 months, the manure were stored for more than six months, the purchased cattle were over 50 heads heads and the stock density were over 3 head/km^−2^, the effect of all these factors would still be less than the combination of the silage fed and badgers.

Based on the presented simulations in this paper, the most important risk factors for NHI in the high-risk areas of England are identified as age, badger densities and silage. Similarly, the most important risk factors for the edge-risk areas are silage and stock densities and for the low-risk areas is the purchased cattle. Thus, a better surveillance of wildlife, especially badgers, in the high-risk areas of England is required to reduce the risk for transmission to cattle, in England and also the whole GB since wildlife epidemio-surveillance is poorly implemented.

## 5. Discussions and the Lessons Learned

The main aim of the current policy is to slow down and stop the spread of bTB into new areas or low-risk areas and achieve a sustained and steady reduction in the disease in high-risk areas. However, bTB remains of great concern in England since policy interventions are not improving the incidence of the disease. Key known risk factors at herd and region levels are reviewed in this paper. Some of the risk factors may have a major impact in some areas of the England but may not be significant elsewhere, due to the different control programs applied and specific epidemiological situations. For illustration, the routine surveillance test intervals in high-risk areas are every year, in edge-risk areas are every two years and in low-risk areas are repeated every four years. Moreover, the wildlife which plays a great role in the spreading of the bTB infection is also dissimilar in each risk areas. However, in all risk areas, farm management was identified as a risk factor but also purchase, wildlife and environment.

Our proposed method is able to identify, not only the probability of the emerging factors which increase the risk of NHI in different risk areas in England individually, but also the strength of the different combinations of those risk factors. In this study, for each risk area, the risk factors with the highest probabilities on NHI were identified as the sensitivity set. Then, the different combination of those risk factors were studied. The results demonstrate that:The sensitivity sets for different risk areas are different.The effect of one risk factor when it is integrated with other risk factors may change significantly. As the results show, in the high-risk areas, the age of cattle is not very concerning on its own; however, when it is associated with the types of feeding (e.g., silage), their joint probability increases the chance of NHI significantly. Furthermore, since the trade of live cattle between different areas is constantly increasing, the risk of moving infected cattle from an OTW region into an OTF area exists. The risk related to the movement of live animals must be emphasised mostly in the low-risk areas. It is thus necessary to remind the people involved in cattle trading of the importance of testing at purchase, especially if the animal introduced in an OTF area comes from a non-OTF-region. Besides, pre-movement testing is important. Thus, limiting movement of cattle onto the farm or purchasing from high-risk areas, along with follow up tests and longer isolation periods (more than 30 days) could reduce the risk of breakdowns. Based on available data, there are five categories from which farmers decide to isolate their newly purchased animals or milking equipment:–7 day–8–15 days–16–21 days–22–30 days–More than 30 daysMost farmers adopt 15 days of isolation because of lack of space. The level of infection in the environment is predicted to decay with a half-life of 34 days (credible intervals 20–71 days). Although decay is rapid, infection may arise months after infectious cattle are removed. Farmers who choose a longer period of isolation have less breakdown incidents.The results are presented graphically and are easy to understand which is one of the strengths of BN.The presented results are not based on one dataset analysis. In fact, the model considered monthly data from 2008 to 2015. Therefore, the model is dynamic, rather than static and, anytime that new evidence is added to the model, the results are updated accordingly. Hence, it is possible to track the effect of risk factors on NHI for the whole period of study.

Moreover, improved farm biosecurity is also required in order to reduce bTB incidence at the herd level: e.g., secure feed storage, secure feeding habits and correct hygiene, as well as a grazing system minimising the cattle/wildlife interface. Thus, for our further work, we should target several points or particular risk factors discussed in this paper but also focus on the role of the environment on bTB transmission as well as determining the exact role of badgers potentially implicated in the transmission of bTB. Further studies are needed to better understand the transmission mechanisms between probable wild maintenance hosts and cattle. The role of environmental persistence of M. bovis needs further investigation in order to assess its role in the epidemiology and transmission of bTB to cattle. Genetic resistance in cattle and wildlife species are also areas that could be targeted for the further research.

### 5.1. Advantages of BNs

BNs, in addition to their simple causal graphical structure, have some other appealing properties:Initial beliefs about the values of each variable (prior probabilities) can be updated in the face of new evidence via Bayes’ theorem.A BN can perform three types of inference: deductive (top-down, forward and predictive), diagnostic (bottom-up, inverse and explanatory) and intercausal (bi-directional “explaining away”). This is useful since the same BNs can be used for both the assessment and the evaluation.Analysts can make probability judgments consistent with the direction of causality.Evidence can be entered into the model, and the effect on the other variables (nodes) can be observed (improving or worsening, and by what magnitude).BNs are able to work with data of different types and sources: they handle a mix of subjective and objective data and, hence, supplement traditional experimental and statistical methods.

### 5.2. Assumptions and Limitations

Various assumptions are made in the DBN model which are a consequence of either lack of complete knowledge on how bTB infection really works and/or poor data (uncertainty about the accuracy, provenance and method of collection are other features of the data). The assumptions made are listed below.

**Assumption** **1.**
*Risk factors are orthogonal, meaning there are no interactions between risk factors.*


**Assumption** **2.**
*The effect of the risk factors on infection rates is additive.*


**Assumption** **3.**
*The model is a reduced order model, meaning it is parsimonious and that variables were removed.*


**Assumption** **4.**
*Spatial factors are not relevant (we created a non-spatial model). If we want to track the movement of badgers, we need empirical data about their patterns of movement and we would use a different modelling method.*


**Assumption** **5.**
*The three risk areas are discrete/separate, i.e., there are no ’grey’ areas in between. However, in the real world, there may or may not be real borders (e.g., rivers) between the areas.*


The model could be developed to address some of the above assumptions. This is likely to result in longer computational times and more resources to collect data and theories on infection risk mechanisms. The current model represents a satisfying position that replicates bTB infection experience and evaluates extant policy.

### 5.3. Software and Toolboxes

In this research, the probabilistic data-driven models were implemented and simulated in BNT version 5.2 in MATLAB^®^ environment (versions 5.0 and 5.1 have a memory leak which seems to sometimes crash BNT).

## 6. Conclusions

The model presented in this paper is able to replicate corresponding new herd incidents from 2008 to 2015. The model demonstrates the influence of particular risk factors upon the risk of breakdown in cattle farms. The model shows the importance of risk factors already considered under current management recommendations which are part of bTB policy aimed at reducing the number of breakdowns and NHI in England’s cattle farms. Our results for high-risk and edge-risk areas suggest that biosecurity is a key risk factor that requires improved control. This is because the mechanisms of disease transmission from badgers to cattle are likely to involve cattle foodstuffs and/or environmental contamination from M. bovis in urine, faeces or sputum. Controls could involve consideration of the type and length of storage of food, as well as limiting contact between cattle and infected badgers sources by fencing, building maintenance and design. Regarding low-risk areas, acquired cattle are the source of the majority of breakdowns in herds, previously free of disease. Pre-movement testing, which was recently introduced to prevent the spread of bTB infection from purchased cattle, can reduce the risk of breakdowns in the low-risk areas.

Finally, it is concluded that applying the DBN modelling approach decreases endogenous uncertainty, reduces investment and policy risks associated with cattle farms husbandry interventions within three risk areas of England and can act as a decision support tool for some of the private, public and community sector stakeholders operating in this field, especially key decision makers and policymakers.

## Figures and Tables

**Figure 1 ijerph-18-03451-f001:**
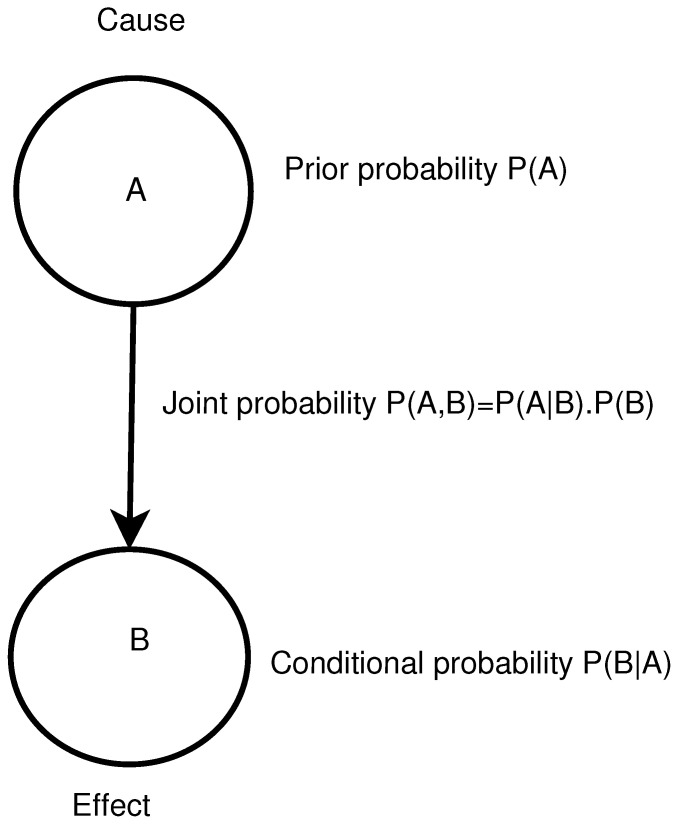
Cause–effect relationships. An outgoing edge from node A to B indicates a causal relation between these two nodes, in which the value of B is dependent on the value of A. A is the parent node of B and B is a child node of A. A conditional probability attached to node B is conditioned on the set of all parents of node B, pa(B), and is represented by P(B|pa(B)).

**Figure 2 ijerph-18-03451-f002:**
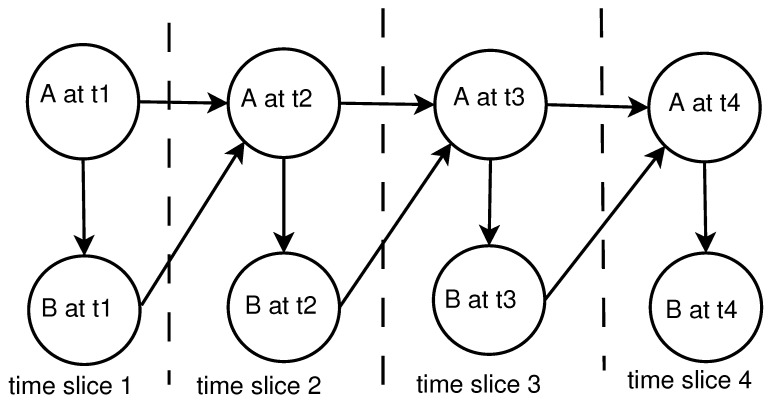
Dynamic Bayesian network representing a feedback loop.

**Figure 3 ijerph-18-03451-f003:**
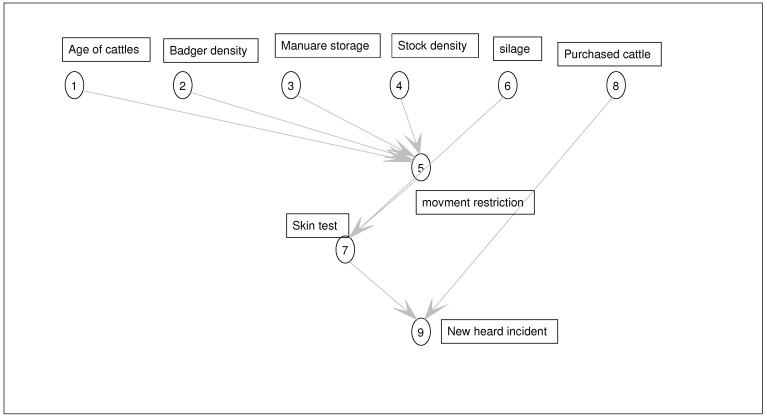
Static Bayesian network model. This network is used to represent the probabilistic relationships between root causes or risk factors and symptoms or evidences. Given evidence about new herd incidents (NHI), the network can compute the probabilities of the strength of various root causes of bTB.

**Figure 4 ijerph-18-03451-f004:**
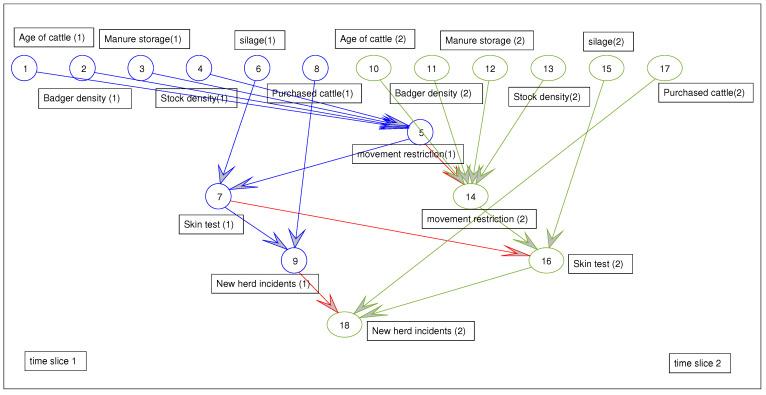
Dynamic Bayesian network model. In this figure, cycles over time can be represented for Figure 3, using an underlying acyclic DBN. In this DBN, skin test results at time t influences the skin test results at time t + 1.

**Figure 5 ijerph-18-03451-f005:**
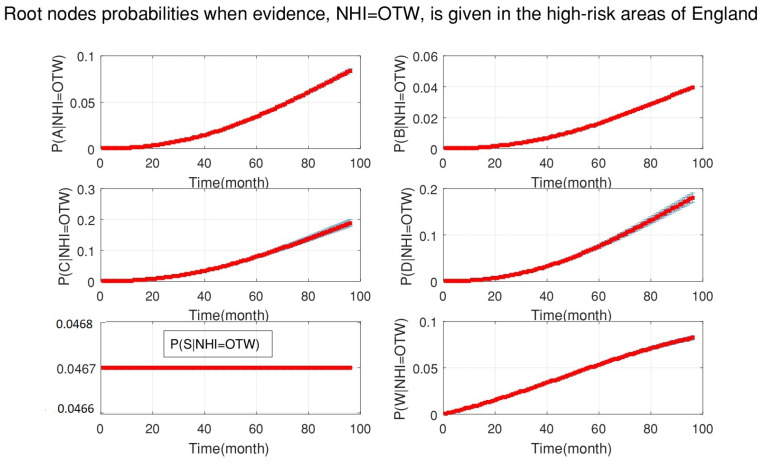
The probabilities of all root nodes (A is the age of cattle, B is the badger density, C is the manure storage, D is the stock density, S is the silage fed and W is the number of purchased cattle) when the evidence, officially TB free status withdrawn (OTW), is added to the system continuously from 2008 to 2015 (bottom-up inference for diagnosis).

**Figure 6 ijerph-18-03451-f006:**
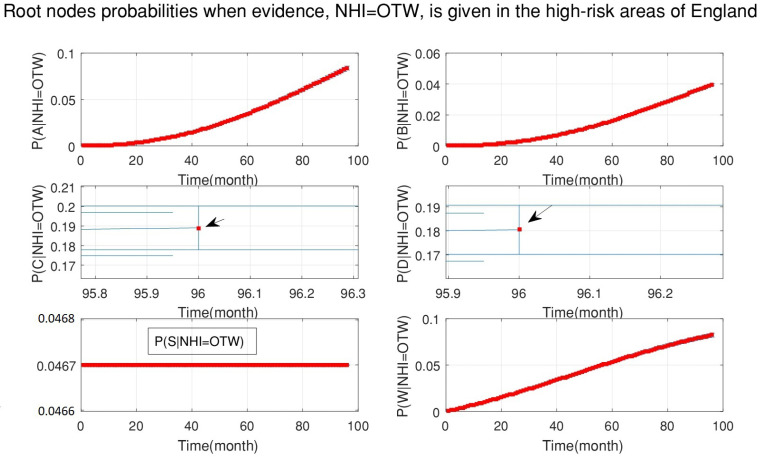
The probabilities of all root nodes (A is the age of cattle, B is the badger density, C is the manure storage, D is the stock density, S is the silage fed and W is the number of purchased cattle) when the evidence, officially TB free status withdrawn (OTW), is added to the system continuously from 2008 to 2015 (bottom-up inference for diagnosis). This figure is a copy of Figure 5 with a clearer view of the middle plots. The middle plots indicate that the probability of manure storage is higher than stock density.

**Figure 7 ijerph-18-03451-f007:**
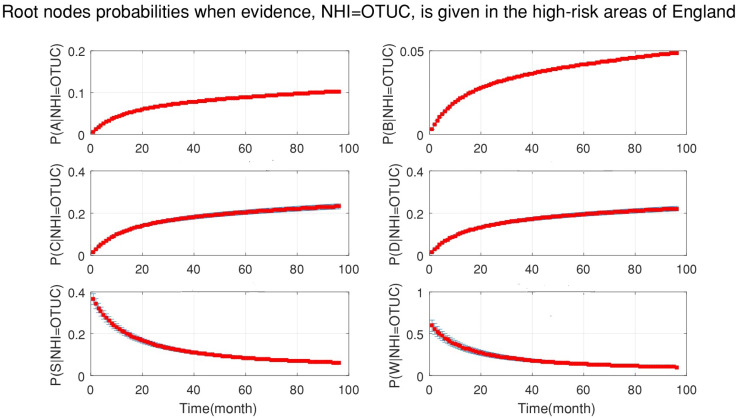
The probabilities of all the root nodes (A is the age of cattle, B is the badger density, C is the manure storage, D is the stock density, S is the silage fed and W is the number of purchased cattle) when new evidence which is officially TB free status unclassified (OTUC) is added to the network continuously from 2008 to 2015 (bottom-up inference for diagnosis).

**Figure 8 ijerph-18-03451-f008:**
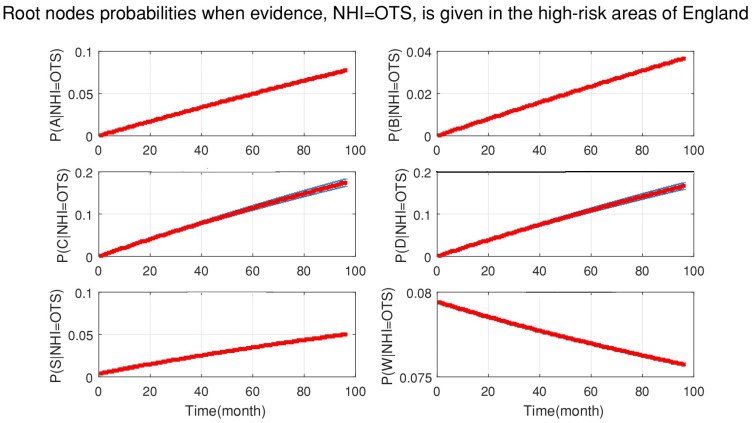
The probabilities of all the root nodes (A is the age of cattle, B is the badger density, C is the manure storage, D is the stock density, S is the silage fed and W is the number of purchased cattle) when new evidence which is officially TB free status suspended (OTS) is added to the network continuously from 2008 to 2015 (bottom-up inference for diagnosis).

**Figure 9 ijerph-18-03451-f009:**
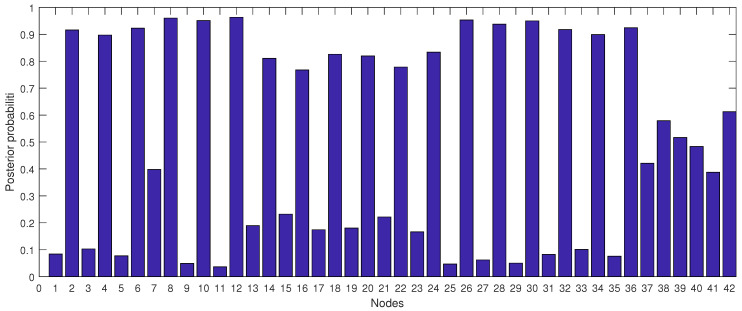
The posterior probabilities of all nodes where bars 1 to 42 represents the posterior probabilities of nodes A>24 m (OTW), A≤24 m (OTW), A>24 m (OTUS), A≤24 m (OTUS), A>24 m (OTS), A≤24 m (OTS), B>3 setts/km${}^{-2}$ (OTW), B≤3 setts/km${}^{-2}$ (OTW), B>3 setts/km${}^{-2}$(OTUC), B≤3 setts/km${}^{-2}$ (OTUC), B>3 setts/km${}^{-2}$ (OTS), B≤3 setts/km${}^{-2}$ (OTS), C>6 months (OTW), C≤6 months (OTW), C>6 months (OTUC), C≤6 months (OTUC), C>6 months (OTS), C≤ 6 months (OTS), D>3 heads/ha (OTW), D≤3 heads/ha (OTW), D>3 heads/ha (OTUC), D≤3 heads/ha (OTUC), D>3 heads/ha (OTS), D≤3 heads/ha (OTS), S:True (OTW), S:False (OTW), S:True (OTUC), S:False (OTUC), S:True (OTS), S:False (OTS), W>50 heads (OTW), W≤50 heads (OTW), W>50 heads (OTUC), W≤50 heads (OTUC), W>50 heads (OTS), W≤50 heads (OTS), F: True (OTW), F: False (OTW), F: True (OTUC), F: False (OTUC), F: True (OTS) and F: False (OTS), respectively.

**Figure 10 ijerph-18-03451-f010:**
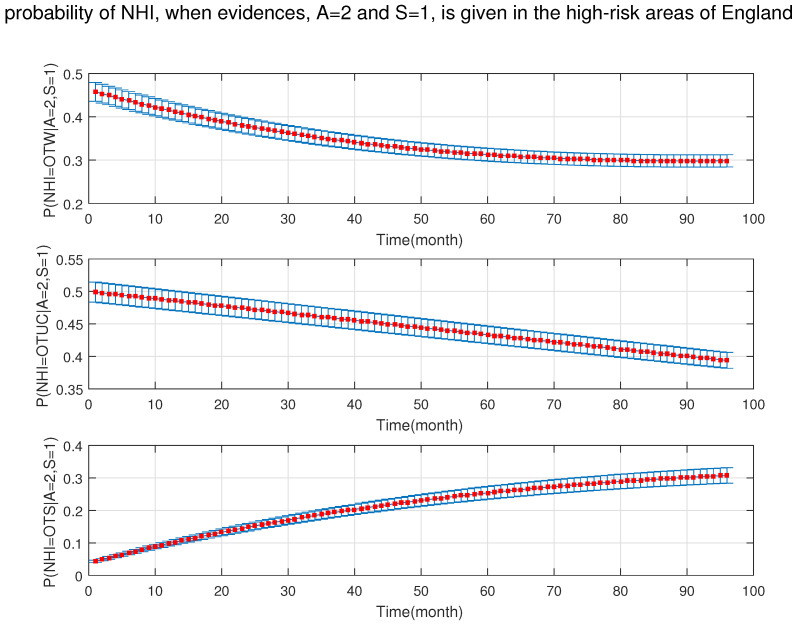
The probability of new herd incidents (officially TB free status withdrawn (OTW), officially TB free status unclassified (OTUC), officially TB free status suspended (OTS)) in high-risk areas of England for A: young cattle (≤24 months old); and S: silage fed.

**Figure 11 ijerph-18-03451-f011:**
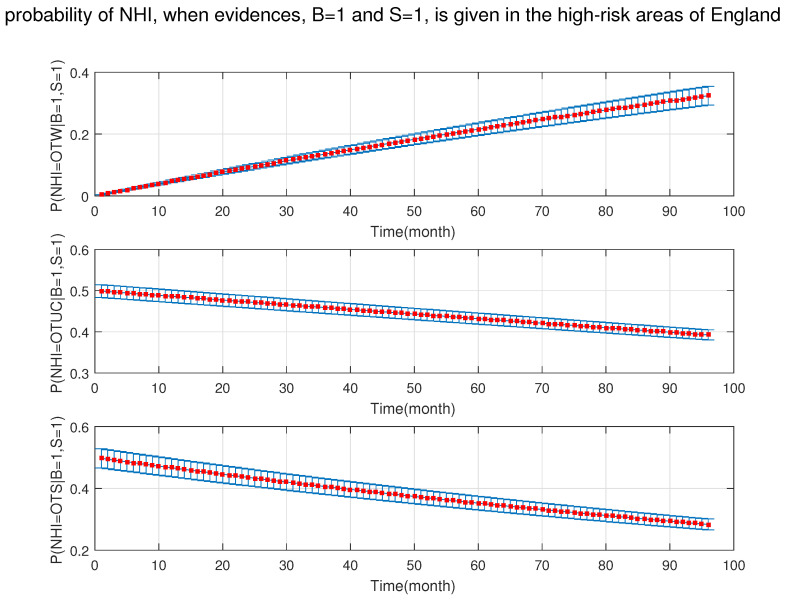
The probability of new herd incident (NHI) (officially TB free status withdrawn (OTW), officially TB free status unclassified (OTUC), officially TB free status suspended (OTS)) in high-risk areas of England where B: badgers density is (>3 setts/ha); and S: silage fed.

**Figure 12 ijerph-18-03451-f012:**
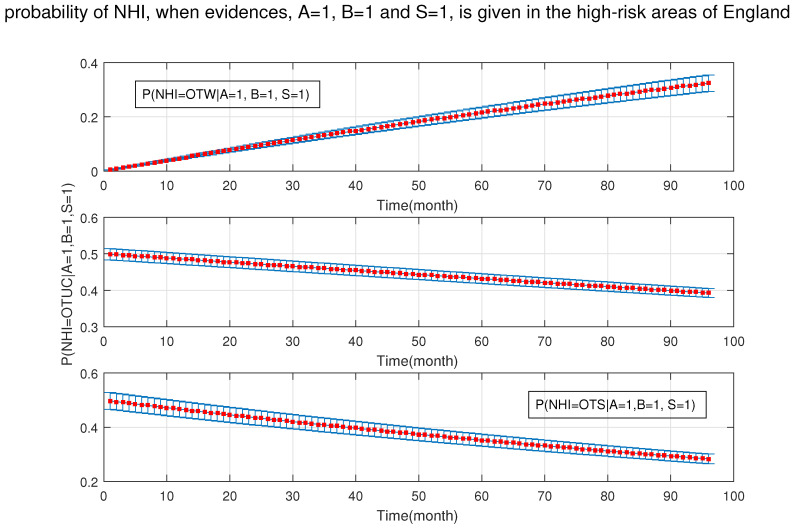
The probability of new herd incident (NHI) (officially TB free status withdrawn (OTW), officially TB free status unclassified (OTUC), officially TB free status suspended (OTS)) in high-risk areas of England where A: cattle are over 24 months, B: badgers density is (>3 setts/ha); and S: silage fed.

**Figure 13 ijerph-18-03451-f013:**
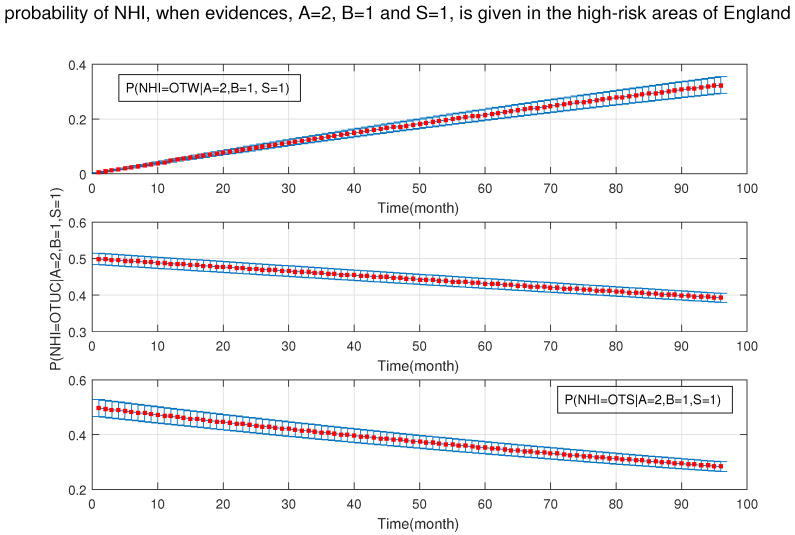
The probability of new herd incident (NHI) (officially TB free status withdrawn (OTW), officially TB free status unclassified (OTUC), officially TB free status suspended (OTS)) in high-risk areas of England for A: cattle ≤24 months, where B: badgers density is >3 setts/ ha; and S: silage fed.

**Figure 14 ijerph-18-03451-f014:**
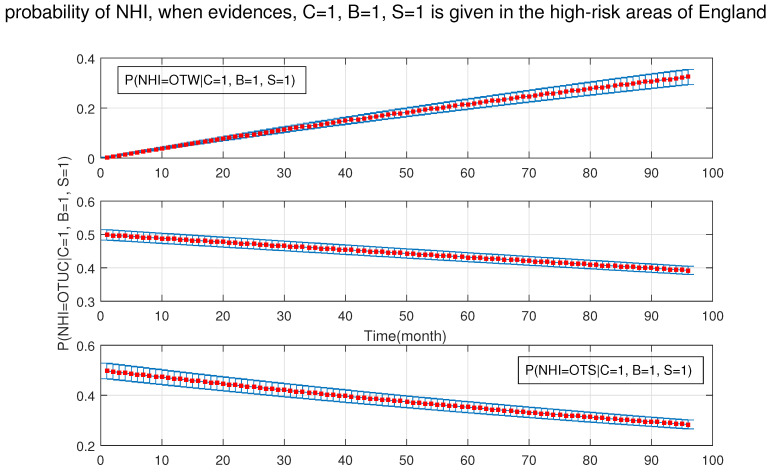
The probability of new herd incident (NHI) (officially TB free status withdrawn (OTW), officially TB free status unclassified (OTUC), officially TB free status suspended (OTS)) in high-risk areas of England where B: badgers density is >3 setts/ha; S: silage fed; and C: manure stored for more than six months.

**Figure 15 ijerph-18-03451-f015:**
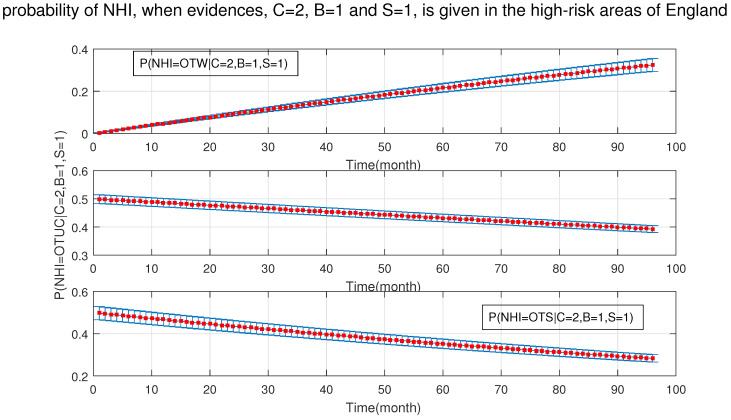
The probability of new herd incident (NHI) (officially TB free status withdrawn (OTW), officially TB free status unclassified (OTUC), officially TB free status suspended (OTS)) in high-risk areas of England where B: badgers density is >3 setts/ ha; S: silage fed; and C: manure stored for less than six months.

**Figure 16 ijerph-18-03451-f016:**
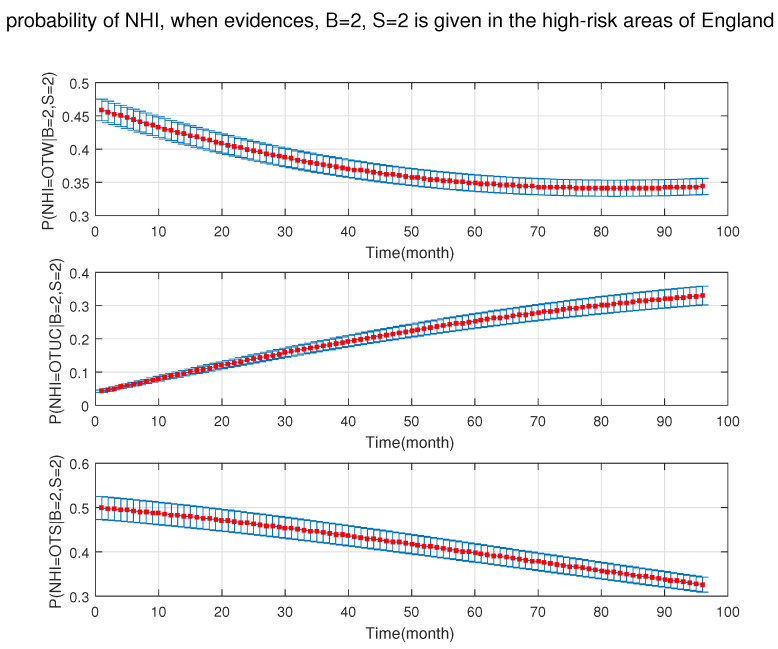
The probability of new herd incident (NHI) (officially TB free status withdrawn (OTW), officially TB free status unclassified (OTUC), officially TB free status suspended (OTS)) in high-risk areas of England where B: badgers density is ≤3 setts/ ha; S: there is no silage fed in the presence of the rest of risk factors (A: cattle are over 24 months; C: manure stored for more than six months; D: stock density is more than 3 head/km${}^{-2}$; and W: the number of purchased cattle is more than 50 heads.

**Figure 17 ijerph-18-03451-f017:**
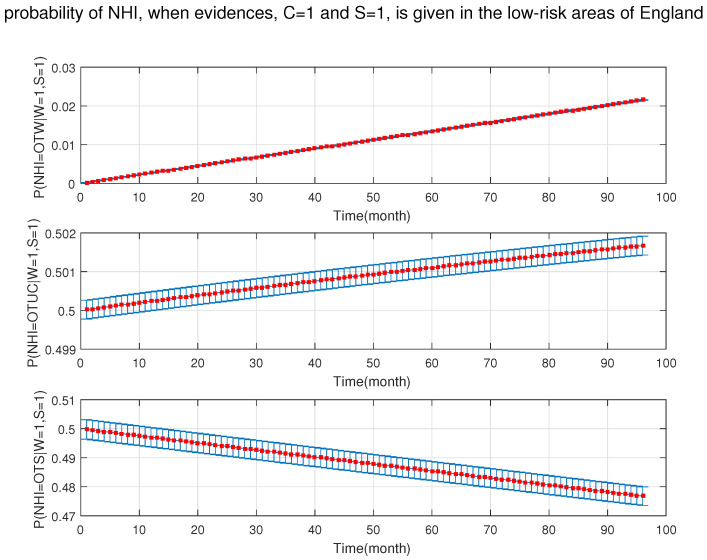
The probability of new herd incident (NHI) (officially TB free status withdrawn (OTW), officially TB free status unclassified (OTUC), officially TB free status suspended (OTS)) in low-risk areas of England where C: manure storage over 6 months; S: there is silage fed in the presence of the rest of risk factors (A: cattle are over 24 months; B: Badger density ≤3 heads/ha; D: stock density is more than 3 head/km${}^{-2}$; and W: the number of purchased cattle is more than 50 heads).

**Figure 18 ijerph-18-03451-f018:**
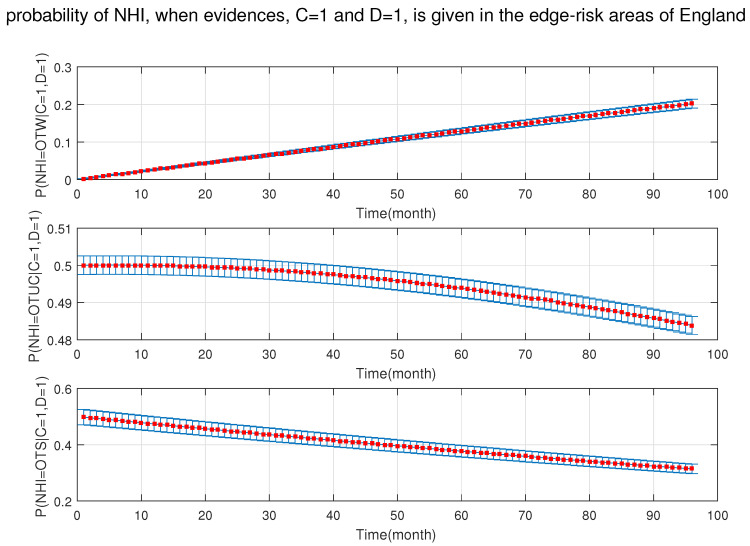
The probability of new herd incident (NHI) (officially TB free status withdrawn (OTW), officially TB free status unclassified (OTUC), officially TB free status suspended (OTS)) in edge-risk areas of England where C: manure storage over 6 months; and D: stock density is more than 3 head/km${}^{-2}$ in the presence of the rest of risk factors (A: cattle are over 24 months; B: Badger density ≤3 heads/ha; S: silage fed; and W: the number of purchased cattle is more than 50 heads).

**Table 1 ijerph-18-03451-t001:** Nodes and their states.

Nodes	State 1	State 2
A: Age of Cattle	>24 months	≤24 months
B: Badgers density	>3 setts/km${}^{-2}$	≤3 setts/km${}^{-2}$
C: Manure storage	>6 months	≤6 months
D: Stock density	>3 heads/ha	≤3 heads/ha
S: Silage	True	False
W: Purchased cattle	>50 heads	≤50 heads
F: Movement restriction	True	False

**Table 2 ijerph-18-03451-t002:** Nodes and their states where NHI indicates new herd incidents, OTW indicates officially TB-free status withdrawn, OTUS indicates officially TB-free status unclassified and OTS indicates officially TB-free status suspended.

Nodes	State 1	State 2	State 3
E: Skin test results	Positive	Inconclusive	Negative
NHI	OTW	OTUC	OTS

**Table 3 ijerph-18-03451-t003:** Prior and posterior probabilities for node A in high-risk areas of England cattle farms.

Nodes	Prior Probability (Lower Bound)	Prior Probability (Estimated)	Prior Probability (Higher Bound)	Posterior Probability (Lower Bound)	Posterior Probability (Estimated)	Posterior Probability (Higher Bound)	Evidence: New Herd Incidents (NHI)
A>24 m	$2.25\times 10^{-9}$	$7.3\times 10^{-6}$	$2.19\times 10^{-6}$	$2.24\times 10^{-5}$	0.08	0.017	OTW
A≤24 m	1	1	1	1	0.916	0.98	OTW
A>24 m	$3.31\times 10^{-5}$	0.0066	$7.59\times 10^{-4}$	0.0016	0.1029	0.0080	OTUS
A≤24 m	1	0.9934	0.9992	0.9984	0.8971	0.9920	OTUS
A>24 m	$4.18\times 10^{-6}$	$8.7\times 10^{-4}$	$1.04\times 10^{-4}$	$3.99\times 10^{-4}$	0.0772	0.0023	OTS
A≤24 m	1	0.9991	0.999	0.9996	0.9228	0.9977	OTS

**Table 4 ijerph-18-03451-t004:** Prior and posterior probabilities for node B in high-risk areas of England cattle farms.

Nodes	Prior Probability (Lower Bound)	Prior Probability (Estimated)	Prior Probability (Higher Bound)	Posterior Probability (Lower Bound)	Posterior Probability (Estimated)	Posterior Probability (Higher Bound)	Evidence: New Herd Incidents (NHI)
B>3 setts/km${}^{-2}$	$5.57\times 10^{-8}$	$3.4\times 10^{-6}$	$1.97\times 10^{-5}$	$5.53\times 10^{-4}$	0.398	0.1533	OTW
B≤3 setts/km${}^{-2}$	1	1	1	0.9994	0.9602	0.8467	OTW
B>3 setts/km${}^{-2}$	$8.21\times 10^{-4}$	0.0031	0.0068	0.0382	0.0488	0.0691	OTUS
B≤3 setts/km${}^{-2}$	0.9992	0.9969	0.9932	0.9618	0.9512	0.9309	OTUS
B>3 setts/km${}^{-2}$	$1.03\times 10^{-4}$	4.0727	$9.37\times 10^{-4}$	0.0098	0.0366	0.0202	OTS
B≤3 setts/km${}^{-2}$	0.9999	0.9996	0.9991	0.9902	0.9634	0.9798	OTS

**Table 5 ijerph-18-03451-t005:** Prior and posterior probabilities for node C in high-risk areas of England cattle farms.

Nodes	Prior Probability (Lower Bound)	Prior Probability (Estimated)	Prior Probability (Higher Bound)	Posterior Probability (Lower Bound)	Posterior Probability (Estimated)	Posterior Probability (Higher Bound)	Evidence: New Herd Incidents (NHI)
C>6 months	$2.92\times 10^{-8}$	$1.74\times 10^{-5}$	$6.58\times 10^{-6}$	$2.91\times 10^{-4}$	0.1891	0.0526	OTW
C≤6 months	1	1	1	0.9997	0.8109	0.9474	OTW
C>6 months	$4.31\times 10^{-4}$	0.0157	0.0023	0.0201	0.2321	0.0237	OTUC
C≤6 months	0.9996	0.9843	0.9977	0.9799	0.7679	0.9763	OTUC
C>6 months	$5.43\times 10^{-5}$	0.0021	$3.12\times 10^{-4}$	0.0052	0.1741	0.0069	OTS
C≤6 months	0.99997	0.9979	0.9997	0.9948	0.8259	0.9931	OTS

**Table 6 ijerph-18-03451-t006:** Prior and posterior probabilities for node D in high-risk areas of England cattle farms.

Nodes	Prior Probability (Lower Bound)	Prior Probability (Estimated)	Prior Probability (Higher Bound)	Posterior Probability (Lower Bound)	Posterior Probability (Estimated)	Posterior Probability (Higher Bound)	Evidence: New Herd Incidents (NHI)
D>3 heads/ha	$2.81\times 10^{-7}$	$1.65\times 10^{-5}$	$8.75\times 10^{-5}$	0.0027	0.1804	0.5857	OTW
D≤3 heads/ha	1	1	0.9999	0.9973	0.8196	0.4094	OTW
D>3 heads/ha	0.0044	0.0149	0.0303	0.1894	0.2216	0.2663	OTUC
D≤3 heads/ha	0.9959	0.9851	0.9697	0.8106	0.7784	0.7337	OTUC
D>3 heads/ha	$5.22\times 10^{-4}$	0.0020	0.0042	0.0487	0.1661	0.0778	OTS
D≤3 heads/ha	0.9995	0.9980	0.9958	0.9513	0.8339	0.9222	OTS

**Table 7 ijerph-18-03451-t007:** Prior and posterior probabilities for node S in high-risk areas of England cattle farms.

Nodes	Prior Probability (Lower Bound)	Prior Probability (Estimated)	Prior Probability (Higher Bound)	Posterior Probability (Lower Bound)	Posterior Probability (Estimated)	Posterior Probability (Higher Bound)	Evidence: New Herd Incidents (NHI)
S:True	0.0467	0.0467	0.0467	0.0467	0.0467	0.0467	OTW
S:False	0.9533	0.9533	0.9533	0.9533	0.9533	0.9533	OTW
S:True	0.3875	0.3660	0.3545	0.1887	0.0621	0.0390	OTUC
S:False	0.6125	0.6340	0.6455	0.8113	0.9379	0.9610	OTUC
S:True	0.0038	0.0043	0.0062	0.0091	0.0501	0.7675	OTS
S:False	0.9962	0.9957	0.9938	0.9909	0.9499	0.2325	OTS

**Table 8 ijerph-18-03451-t008:** Prior and posterior probabilities for node W in high-risk areas of England cattle farms.

Nodes	Prior Probability (Lower Bound)	Prior Probability (Estimated)	Prior Probability (Higher Bound)	Posterior Probability (Lower Bound)	Posterior Probability (Estimated)	Posterior Probability (Higher Bound)	Evidence: New Herd Incidents (NHI)
W>50 heads	$4.28\times 10^{-5}$	$6.63\times 10^{-4}$	0.0017	0.0044	0.0822	0.1408	OTW
W≤50 heads	1	0.9993	0.9983	0.9956	0.9178	0.8592	OTW
W>50 heads	0.6306	0.5956	0.5769	0.3071	0.1010	0.0635	OTUC
W≤50 heads	0.3694	0.4044	0.4231	0.6929	0.8990	0.9365	OTUC
W>50 heads	0.0794	0.0794	0.0792	0.0790	0.0757	0.0185	OTS
W≤50 heads	0.9206	0.9206	0.9208	0.9210	0.9243	0.9815	OTS

**Table 9 ijerph-18-03451-t009:** Prior and posterior probabilities for node F in high-risk areas of England cattle farms.

Nodes	Prior Probability (Lower Bound)	Prior Probability (Estimated)	Prior Probability (Higher Bound)	Posterior Probability (Lower Bound)	Posterior Probability (Estimated)	Posterior Probability (Higher Bound)	Evidence: New Herd Incidents (NHI)
F: True	$3.68\times 10^{-7}$	$4.46\times 10^{-5}$	$1.15\times 10^{-4}$	0.0036	0.4209	0.7386	OTW
F: False	1	1	0.9999	0.9964	0.5791	0.2614	OTW
F: True	0.0054	0.0401	0.0401	0.2463	0.5168	0.3331	OTUC
F: False	0.9946	0.9599	0.9599	0.7537	0.4832	0.6669	OTUC
F: True	$6.84\times 10^{-4}$	0.0053	0.0055	0.0633	0.3875	0.0973	OTS
F: False	0.9993	0.9947	0.9945	0.9367	0.6125	0.9027	OTS

**Table 10 ijerph-18-03451-t010:** Prior and posterior probabilities for node E in high-risk areas of England cattle farms.

Nodes	Prior Probability (Lower Bound)	Prior Probability (Estimated)	Prior Probability (Higher Bound)	Posterior Probability (Lower Bound)	Posterior Probability (Estimated)	Posterior Probability (Higher Bound)	Evidence: New Herd Incidents (NHI)
E: Positive	$3.68\times 10^{-7}$	$4.46\times 10^{-5}$	$1.15\times 10^{-4}$	0.0036	0.4209	0.7386	OTW
E: Inconclusive	0.9533	0.9533	0.9532	0.9499	0.5521	0.2492	OTW
E: Negative	0.0467	0.0467	0.0467	0.0465	0.0270	0.0122	OTW
E: Positive	0.0054	0.0401	0.0401	0.2463	0.5153	0.3331	OTUC
E: Inconclusive	0.6073	0.5958	0.6072	0.5765	0.4459	0.6435	OTUC
E: Negative	0.3872	0.3641	0.3526	0.1772	0.0388	0.0234	OTUC
E: Positive	$6.84\times 10^{-4}$	0.0053	0.0055	0.0633	0.3875	0.0973	OTS
E: Inconclusive	0.9956	0.9906	0.9885	0.9305	0.5805	0.1398	OTS
E: Negative	0.0037	0.0040	0.0060	0.0062	0.0320	0.7630	OTS

**Table 11 ijerph-18-03451-t011:** Prior and posterior probabilities for evidence in high-risk areas of England cattle farms.

Nodes	Prior Probability (Lower Bound)	Prior Probability (Estimated)	Prior Probability (Higher Bound)	Posterior Probability (Lower Bound)	Posterior Probability (Estimated)	Posterior Probability (Higher Bound)
OTW	0.1197	0.1237	0.1238	0.1728	0.4617	0.4704
OTUC	0.8803	0.8763	0.8762	0.8272	0.5383	0.5296
OTS	0.0431	0.0429	0.0429	0.0405	0.0264	0.0259

## Data Availability

Data are available publicly as secondary data in Quarterly TB in cattle in Great Britain statistical notice (data to March 2018): https://www.gov.uk/government/statistical-data-sets/tuberculosis-tb-in-cattle-in-great-britain (accessed on 24 March 2018).

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
