# Peer review of "Evaluating the Bovine Tuberculosis Eradication Mechanism and Its Risk Factors in England’s Cattle Farms"

_ijerph, 2021, doi:10.3390/ijerph18073451_

Round 1

Reviewer 1 Report

This paper presents a dynamic Bayesian network allowing to identify the main risk factors for the appearance of bovine tuberculosis in cattle in England.

The paper is very clear, and well structured. The description of the bayesian model is suitable to the non-specialists of the domain. The results demonstrate the usefulness of this type of statistical tools to identify important risk factors.

I have only a few minor comments:

Lines 208-209: it is mentioned that causal relationships between the different nodes are elicited by experts and the available literature.

Some details about the number of sollicited expert, how they were chosen and how they collaborated (individual opinion, opinion resulting from a consensus, opinion of the majority of them?)  could be added. 

In the same way, some details about the bibliographic search could be added.

The simulation results are presented in the discussion. I think that this part could be presented in the results section.

I cannot see the difference between the figure 5 and 6 legends?

Line 299-300: a better surveillance of wildlife is required to reduce the risk of transmission to cattle, at national but also international levels.

Why the surveillance of wildlife at international level is recommanded since England is an island and the likelihood of badgers from other countries invading the UK is very low.

Reviewer 2 Report

I thoroughly enjoyed the article. The manuscript clear, well-written and summarizes the complex Bayesian Network convincingly and informatively.

You can find additional comments in the attachment.
